# Medically Important Alterations in Transport Function and Trafficking of ABCG2

**DOI:** 10.3390/ijms22062786

**Published:** 2021-03-10

**Authors:** László Homolya

**Affiliations:** Research Centre for Natural Sciences, Institute of Enzymology, H-1117 Budapest, Hungary; homolya.laszlo@ttk.hu; Tel.: +36-1-382-6608

**Keywords:** ABC (ATP-binding cassette) transporters, multidrug resistance, transport, trafficking, urate, mutations, polymorphisms

## Abstract

Several polymorphisms and mutations in the human ABCG2 multidrug transporter result in reduced plasma membrane expression and/or diminished transport function. Since ABCG2 plays a pivotal role in uric acid clearance, its malfunction may lead to hyperuricemia and gout. On the other hand, ABCG2 residing in various barrier tissues is involved in the innate defense mechanisms of the body; thus, genetic alterations in *ABCG2* may modify the absorption, distribution, excretion of potentially toxic endo- and exogenous substances. In turn, this can lead either to altered therapy responses or to drug-related toxic reactions. This paper reviews the various types of mutations and polymorphisms in *ABCG2*, as well as the ways how altered cellular processing, trafficking, and transport activity of the protein can contribute to phenotypic manifestations. In addition, the various methods used for the identification of the impairments in ABCG2 variants and the different approaches to correct these defects are overviewed.

## 1. Introduction

The ABCG2 protein (also named breast cancer resistance protein—BCRP, or mitoxantrone resistance protein—MXR) is a member of the ABC (ATP-binding cassette) protein superfamily. A distinguishing hallmark of ABC proteins is the presence of Walker A, Walker B, and the so-called ABC signature (typically LSGGQ) motifs in their sequences. The members of this large protein family are present in all living organisms, ranging from prokaryotes through fungi, plants, invertebrates to vertebrates. The design of ATP-binding fold and its connection to transport mechanisms seem evolutionarily beneficial, as they have been conserved through evolution [1]. In the human genome, there are 48 genes encoding ABC proteins, which are classified into seven subfamilies (denoted from A to G) primarily on the basis of sequence homology. Since an increasing number of structural and functional data are available for ABC proteins, a new classification based on these parameters has recently been proposed [2]. Most of the human ABC proteins are membrane proteins mediating translocation of substances across biological membranes using the energy of ATP binding and hydrolysis. There are some peculiar members of the family, like the regulatory ABC proteins, exemplified by the sulfonylurea receptors (SUR1/ABCC8 and SUR2/ABCC9), which control the function of other membrane proteins; or the cystic fibrosis transmembrane regulator (CFTR/ABCC7), which is an ion channel facilitating downhill chloride transport across the membrane.

Some human ABC proteins are specialized in the transport of one or a limited number of substrates. For example, MDR3/ABCB4 mediates phosphatidylcholine transport in the canalicular membrane of hepatocytes. In contrast, MDR1 (P-glycoprotein, ABCB1) is rather promiscuous, transporting a large variety of unrelated molecules. Membrane transporter proteins with broad substrate recognition may confer resistance in cells to multiple drugs, i.e., causing cross-resistance in tumor cells. These transporters are called multidrug resistance (MDR) proteins, although they also play a pivotal role at important physiological tissue barriers controlling the uptake and excretion of endo- and xenobiotics. In humans, there are multidrug transporters from the ABCB, the ABCC, and the ABCG subfamilies. These MDR proteins with their broad and partially overlapping substrate recognition, as well as with their tissue- and cell type-specific expression constitute a complex physiological network, called the chemoimmunity system, which is an essential part of the innate defense system against harmful substances [3].

ABCG2 was originally identified as a multidrug transporter in multidrug-resistant cancer cell lines, in which none of the two MDR proteins known at that time (MDR1/ABCB1 and MRP1/ABCC1) was expressed [4,5]. In drug-selected cells and certain tumors, ABCG2 is massively overexpressed and may contribute to the poor clinical outcome of these tumors [6,7]. Its physiological presence in normal tissues has also been demonstrated—first in placenta [8], and subsequently in a large variety of other tissues. The wide-ranging but still specific tissue distribution, combined with the broad substrate recognition, makes ABCG2 an essential element of the chemoimmunity network. This paper will provide an overview of the current knowledge of the structure and function of the ABCG2 protein, as well as on the medical conditions when the transporter improperly operates. A special focus will be put on the naturally occurring mutations and polymorphisms in ABCG2, which cause diminished transport activity and/or impaired cellular routing. Finally, the various methods assessing the function and trafficking of ABCG2, as well as the different efforts to rescue detrimental phenotypes caused by the faulty transporter will be discussed.

## 2. Architecture of ABCG2

ABCG2 belongs to the ABCG subfamily, the members of which, in addition to sequence homology, exhibit considerable structural similarities. To our recent knowledge, the minimal structure of a functional ABC transporter is composed of two cytoplasmic nucleotide-binding domains (NBDs) and two transmembrane domains (TMDs). The usual arrangement of domains in the core protein is as follows: TMD1-NBD1-TMD2-NBD2. Contrary to the canonical ABC transporters, members of the ABCG subfamily possess only one NBD and one TMD; thus, they are called half-transporters. Moreover, the domain order in ABCG proteins is reversed, i.e., NBD is localized N-terminally to TMD. This reverse domain arrangement could be one of reasons for the sensitivity of ABCG2 to tagging at the C-terminus [9], contrary to many other ABC transporters, which can regularly be tagged C-terminally. To form a functional complex, ABCG proteins, like other ABC half-transporters, assemble into either homo- or heterodimers. While ABCG2 solely forms homodimers, ABCG5 and ABCG8 are obligate heterodimers. In contrast, two other members of the subfamily, ABCG1 and ABCG4 can form both homo- and heterodimers [10].

Membrane topology models of ABCG2 suggest six membrane-spanning helices, a relatively short C-terminal tail, and short loops between the transmembrane helices (TMHs) except for the last extracellular loop (EL3) between TMH5 and TMH6 [11]. N-glycosylation on asparagine 596 located in the EL3 loop has also been demonstrated [12,13]. Although initial reports suggested that glycosylation at N596 was not essential for proper expression, localization, and function, subsequent studies demonstrated N-glycosylation to be an important checkpoint determining the stability and intracellular trafficking of the transporter [14,15]. There are twelve cysteine residues in ABCG2, but only three of them are positioned in an oxidative milieu, and thus capable of forming disulfide bonds. All three of these cysteines are located in the EL3 loop, and while an intramolecular disulfide bond is established between C592 and C608, an intermolecular disulfide bridge is formed between the two halves of the homodimer at C603. Whereas the latter disulfide bond is not required for proper trafficking and function of ABCG2 [16,17,18], the C592-C608 intramolecular disulfide bond represents another critical checkpoint for protein folding and trafficking [15,18].

Although the X-ray structures of isolated NBDs have been available since the late nineties, the first high-resolution structures of full-length ABC transporters were only published in 2006 and 2007 [19,20]. In the last decade, the spread of cryogenic electron microscopy (cryo-EM) and the substantial progress in crystallography have given a boost to our understanding of ABC protein structures. However, homology modeling was not quite applicable to ABCG proteins, as the members of this family are rather distinct from other ‘classical’ ABC transporters, such as the P-glycoprotein (MDR1/ABCB1) or the CFTR/ABCC7. The appearance of the first high-resolution structure of an ABCG protein, i.e., that of the heterodimeric sterol transporter ABCG5/ABCG8, was therefore a breakthrough [21], fueling extensive homology modeling of ABCG2 [22,23,24]. Subsequently, several ABCG2 structures based on cryo-EM analyses have been published [25,26,27,28]. The structural characteristics of ACBG2, obtained from these cryo-EM studies and from parallel molecular dynamic stimulations, are recently reviewed in [29,30].

There are some distinguishing structural features in the ACBG2 as compared to the full-length ABC transporters. In general, the structure of ABCG2 is more compact, the NBDs are positioned close to the TMDs. A similar compact arrangement was observed for the ABCG5/ABCG8 crystal structure [21], which resembles the architecture of BtuCD-like bacterial importers, rather than that of MDR1-like transporters. This originates from the relatively short transmembrane helices, possessing no cytosolic extension unlike the helices in the classical ABC proteins, which create a sort of spacer between the TMDs and NBDs. In the MDR1-like proteins, two of the elongated four pairs of helices cross over and bind to the opposite NBD, while the other two pairs interact with the ipsilateral NBD. The interfacings to NBDs are realized by small, so-called coupling helices at the cytosolic tip of the elongated TMHs [19]. In contrast, there is only one coupling helix in each half of the ABCG2 dimer (between TMH2 and TMH3), which does not cross over to the other half. However, an amphipathic helix, called the connecting helix, linked to only TMH1 and reclining against the membrane bilayer, provides an additional TMD-NBD interface in ABCG2.

Another distinct feature of the structure of ABCG2, as well as of ABCG5-ABCG5, is the relatively closed conformation in the absence of ATP. In the classical ABC transporters in this ‘apo’ form (without ATP), the NBDs are located far from one another, and consequently, the intracellular parts of TMHs also remain apart, forming a large central cavity, the main substrate-binding pocket, which is widely open to the cytoplasm [19]. The presence of a similar cavity at the cytoplasmic side of ABCG2 (cavity 1) has been reported by cryo-EM studies using an anti-ABCG2 antibody to reduce flexibility in the structure [25,26,27]. Nevertheless, the NBDs, and consequently the intracellular parts of TMHs, are closer to one another in ABCG2 than in MDR-like transporters, resulting in a more compact structure even in the absence of ATP. Residues in this central cavity were shown to be essential not only for transport function but also for biogenesis [24]. Interestingly, a study using no anti-ABCG2 antibodies for structure stabilization reported the lack of cavity 1 [28]. An additional prominent feature in the inward-facing structure (apo form) of ABCG2 is the hydrophobic di-leucine valve (L554 and L555) separating the central substrate-binding pocket from an additional cavity (cavity 2) located toward the extracellular part of ABCG2 [25,26]. Experiments supplemented by molecular dynamic simulations demonstrated an essential role for this di-leucine plug in the transport function [31]. Putting together the different structures in the absence and presence of ATP and/or substrates, MDR1-like proteins seem to alternate between a widely open inward-facing and a fairly open outward-facing conformations, whereas the translocation of substrates through ABCG2 via cavities 1 and 2 rather involves a peristaltic-like movement.

With regard to the NBDs, both sequences and structures are fairly conserved. The two composite ATP-binding pockets are constituted by two separate NBDs in a head-to-tail orientation, i.e., one ATP molecule binds to the Walker A and B motifs of one NBD and to the ABC signature sequence of the other NBD. Unlike in full-length ABC transporters, the cytoplasmic part of the homodimeric ABCG2 is composed of two identical halves, but otherwise the ABC-folds in ABCG2 are structurally similar to that of the classical ABC transporters. It is worth noting that a phenylalanine at position 142 in ABCG2 interacts with the connecting helix, representing a key residue in TMD-NBD interface assembly and a critical checkpoint for protein folding and function [32,33]. Interestingly, this amino acid is analogous to F508 in CFTR/ABCC7, the mutation of which is responsible for diminished trafficking of CFTR, and ultimately the cystic fibrosis (CF) phenotype.

## 3. The Physiological Functions of ABCG2, and Its Role in Multidrug Resistance

### 3.1. The Physiological Roles of ABCG2

As mentioned previously, ABCG2 is overexpressed in drug-resistant cell lines and tumors. Habitually, it is expressed at a relatively high level in cell types located at the entry and exit boundaries of the body, as well as in barrier tissues at the borders of sanctuary sites [34,35]. These include the epithelial cells of the gastrointestinal track, especially in small intestine enterocytes [36], the kidney tubular epithelial cells [37], hepatocytes [34], placental syncytiotrophoblasts [38], mammary alveolar epithelial cells (a part of the blood-milk barrier) [39], and brain capillary endothelial cells (a key element of the blood-brain barrier) [40,41]. In these polarized epithelial and endothelial cells, ABCG2 is localized to the apical plasma membrane domain. In addition to these cells constituting tissue barriers, ABCG2 is also expressed in various types of stem cells including hematopoietic stem cells [42], pluripotent stem cells [43,44], and cancer stem cells [45,46,47,48]. Interestingly, ABCG2 is also present in the membrane of red blood cells (RBCs) [34,49,50].

As is typical of a multidrug transporter, ABCG2 recognizes a vast variety of compounds as transported substrate molecules. These include uric acid in the first place, but also various endogenous conjugated hormones and metabolites, several hydrophobic and amphipathic drugs, as well as their conjugates [51,52,53]. This promiscuity and the tissue distribution detailed above delineate the physiological function of this transporter. In general, ABCG2—depending on its location—restricts the uptake or facilitates the excretion of potentially toxic or unwanted substances. Specifically, in the brain capillaries, ABCG2 restricts the passage of substances through the blood-brain barrier, whereas in the placenta, it protects the fetus from maternally derived toxins. For instance, ABCG2 restricts the maternal-fetal transfer of bile acids, which is especially important in expecting mothers with intrahepatic cholestasis of pregnancy, a frequent liver disease leading to augmented serum levels of bile acids [54,55]. In the small intestine, ABCG2 controls the absorption of various molecules and participates in extra-renal clearance of uric acid; in the kidney proximal tubules, it contributes to the elimination of unwanted toxins and metabolites, including uric acid. Impaired ABCG2-mediated urate transport may lead to gout or hyperuricemia, therefore, specific mutations and polymorphisms in ABCG2 are genetic risk factors for these conditions [37,56,57,58] to be discussed in detail in Section 6. Interestingly, a recent study reported unequal contribution of ABCG2 to renal and extra-renal clearance of uric acid [58]. In mammary alveolar epithelial cells, this transporter influences the milk composition. Endogenous substrates transported by ABCG2 through the blood-milk barrier include riboflavin (vitamin B_2_) and bile acids [59,60]. Certainly, vigilance is required for breast-feeding mothers, as various medications can be transported by ABCG2 into the milk [39,61]. The relevance of the Abcg2-mediated drug transport for the dairy industries is also self-evident [62,63].

In these physiological boundaries, ABCG2 accomplishes this ‘bouncer duty’ in a coordinated fashion together with the other MDR proteins, MDR1/ABCB1 and MRP1/ABCC1, exploiting their partially overlapping substrate recognition and specific subcellular localization. In polarized epithelial cells, ABCG2 is localized to the apical membrane ipsilaterally to MDR1/ABCB1 and contralaterally to MRP1/ABCC1, whereas in cerebral endothelial cells, all three major MDR proteins reside at the same side, i.e., the apical membrane [64]. Accordingly, ABCG2 along with other MDR proteins potentially alters the absorption, distribution, and excretion, as well as, consequently, the metabolism and toxicity (ADME-Tox properties) of pharmaceutical drugs. Especially important is the potential contribution of these transporters to drug–drug interactions, since modification of one (or more) of the MDR proteins by a drug may greatly influence the pharmacokinetics of another one. Therefore, the examination of drug interactions with MDR proteins, including ABCG2, is a requirement in preclinical drug development [53,65,66]. Interestingly, in mammary epithelial cells, the apically localized ABCG2 and the basolateral MRP4/ABCC4 counteract one another in bile acid transport [60].

The physiological role of ABCG2 in red blood cells and stem cells is enigmatic to some extent. Since phototoxic porphyrins, such as the plant-derived pheophoride A and the heme precursor protoporphyrin IX (PPIX), are noted substrates of ABCG2, its expression in the erythroid precursor cells and in mature RBCs may indicate its involvement in heme metabolism [50,67,68]. It is worth noting, however, that numerous membrane proteins without known function in RBCs are present in their membrane, i.e., the sterol transporter ABCA1 (http://rbcc.hegelab.org/, accessed on 9 February 2021) [69,70]. It is plausible that many of these membrane proteins can be just remnants from previous stages of cell differentiation and maturation. In various stem cell types, a protective role similar to that observed at the border of sanctuary sites has been proposed for ABCG2 [42,43,44]. Stem cells are poised between self-renewal and differentiation, and are thus exceptionally sensitive to environmental factors. ABCG2 can contribute to the stem cells’ self-protective mechanisms. The presence of the transporter may, however, backfire in cancer stem cells, as they can provide tumors with drug-resistant cell populations.

### 3.2. The Involment of ABCG2 in Multidrug Resistance of Cancer

Beyond its physiological roles, ABCG2 has been implicated in cancer multidrug resistance (recently reviewed in [7]). A large variety of chemotherapeutic agents has been identified as ABCG2 substrates. First, the anti-cancer drug mitoxantrone has been demonstrated to be exported by ABCG2, thus reducing its intracellular accumulation [4,71,72,73]. Interestingly, a kinetic analysis indicated that mitoxantrone is extruded by ABCG2 not from the cytosol but directly from the plasma membrane, where the drug accumulates [74]. Other anti-cancer drugs identified as ABCG2 substrates include flavopiridol [73,75], methotrexate [76], topotecan, and irinotecan [77,78]. In addition, several prominent tyrosine kinase inhibitors (TKIs) used in chemotherapies, such as gefitinib [79,80,81], imatinib [81,82,83], sunitinib [84], and nilotinib [83,85], were proven to be transported by ABCG2. The anti-cancer agents doxorubicin and daunorubicin have also been reported as ABCG2 substrates [4,72,75], but eventually it was revealed that these drugs are transported only by the R482G ABCG2 variant [3,86].

Expression of ABCG2 in tumors often correlates with poor prognosis, especially in hematopoietic malignancies, such as acute myeloid leukemia [87], but also in solid tumors, including diffuse large B-cell lymphoma [88]. However, clinical data are often conflicting like in the case of acute lymphocytic leukemia [89,90,91], or of breast carcinoma [92,93,94]. Several other studies demonstrated correlation between ABCG2 expression and response to chemotherapy, even to drugs, which are not ABCG2 substrates. These inconsistencies can originate from the modulatory effect of other drug resistance mechanisms, most evidently the presence of other MDR proteins. In addition, the methods employed to determine ABCG2 expression could be dubious, originating from the potential cross-reactivity of applied antibodies, or from the fact that mRNA levels of membrane proteins often do not correlate with the protein levels. The genetic background of patients could give an extra hue to these clinical studies as mutations and polymorphisms may alter the input of ABCG2 into the clinical outcome or the response to various drugs; therefore, proper stratification of patients is crucial for these analyses. In summary, the role of ABCG2 in tumors has been implicated, but its actual contribution to the clinical multidrug resistance is still unclear [6,7].

## 4. Mutations and Polymorphisms in ABCG2

### 4.1. Classifications of Genetic Variants of ABCG2

Normal functioning of ABCG2 can be modulated, attenuated or abolished by mutations or polymorphisms, which in turn may lead to medical conditions. These genetic alterations can affect the transport activity of ABCG2 either by limiting its ATPase activity, or by altering its substrate affinity and/or substrate profile. In addition, when the transporter is not expressed at an adequate level, it cannot fully accomplish its physiological role. Since ABCG2 operates as a pump protein residing in the plasma membrane and expelling various substrates from the cell, not only sufficient expression but also proper cellular localization is a prerequisite for normal function. Diminished cell surface protein level can stem either from reduced overall expression caused by an early stop codon, mRNA instability, protein folding problems, and increased degradation; or from trafficking problems, such as retention in a cellular compartment, halted posttranslational modification, intracellular sequestration, and augmented internalization. The former factors affect general expression of the transporter, whereas trafficking defects alter its steady-state concentration on the cell surface. Based on this, mutations and polymorphisms can be categorized as affecting (i) the general expression level, (ii) the cellular trafficking, or (iii) the transport activity. In many cases, genetic variations alter not only one of these parameters, but various combinations of them.

A classification of different ABCG2 variants has been proposed by Tamuara et al. [95]. This categorization is based on the protein expression level and drug resistance profile of the variants. The four groups are defined as follows: (i) variants with wild type-like drug resistance profile; (ii) mutants with acquired doxorubicin and daurorubicin resistance, as well as with prazosin-stimulated ATPase activity; (iii) non-expressing mutants; (iv) and others possessing normal expression levels but altered drug resistance profile. Recently, a novel and more systematic classification was suggested by Heyes et al. [96]. In this system, the main categories, ranging from one to three, are based on the cell surface expression of the ABCG2 variant: normal (as wild type, wt), reduced, and increased, whereas subcategories denoted by a and *b* indicate whether the transport function is preserved (or even elevated) and reduced, respectively. This classification is simple and reasonable, but does not distinguish between normal variants and gain of function mutants, and does not take into account whether the reduced cell surface appearance is due to general expression problems or trafficking defects despite the fact that rescuing a phenotype caused by one or the other requires distinct interventions.

It is, therefore, worth considering a new classification of ABCG2 mutations similar to that was previously introduced for CFTR variants, which is based on so-called theratypes [97]. In this classification, CFTR variants are categorized according to the nature of their defect and the specific strategies required for phenotype correction. While Class 1 mutations affect protein production, Class 2 mutations impair trafficking. Class 3 and 4 mutations diminish transport function by affecting channel gating and conductance, respectively. Class 5 mutations lead to reduced protein levels, whereas Class 6 mutations reduce plasma membrane half-lives. Finally, Class 7 mutations are the so-called unrescuable genetic variants, e.g., those containing large deletions [97].

With regard to the ABCG2 mutations, we propose here a new classification, which embraces the logic and architecture of the CFTR mutation categorization, but also considers the specific features of ABCG2. Although this classification does not follow the order of sequential cellular event, such as transcription, translation, post-translational modification, trafficking, and degradation, it is rather based on conventions used for CFTR for many years. However, conforming of mutation classifications of various ABC proteins may help to avoid confusion and to adapt interventions to rescue phenotypes caused by the mutations from the same categories (see Section 7).

### 4.2. Class 0 Mutations

Various databases based on genome and transcriptome sequences list an enormous number (over 1000) of mutations and polymorphisms in the *ABCG2* gene. These also include the synonymous mutations and the genetic variants in the non-coding regions, but from a practical point of view, we focus here primarily on non-synonymous single-nucleotide polymorphisms (SNPs) in the coding region. A comprehensive collection of mutations in ABC proteins, including those of ABCG2, is available at http://abcmutations.hegelab.org/ (accessed on 9 February 2021) [98,99]. This database (named ABCMdb) summarizes not only the naturally occurring but also the artificially generated genetic variants, and annotates them with relevant literature data. Inclusion of artificial mutations in such databases is of great help in gaining insights into structure-function relationships and in designing future experiments.

One group of ABCG mutations can be formed from those that do not affect the function, expression, and trafficking considerably (marked as Class 0). A characteristic representative of this group is the frequent missense polymorphism V12M (rs2231137) [49,100]. The minor allele frequency (MAF) of this SNP falls between 0.19 and 0.33 in the Asian populations and is around 0.06 in Europe [101,102]. Other members of this class are K360del and T434M [103,104]. The specific cellular events affected by the mutations of each group are depicted in Figure 1, whereas detailed data on the major representatives of the various SNP categories are provided in Table 1.

### 4.3. Class 1 Mutations

The next group is composed of mutations that impair protein production (Class 1). Among them, Q126X (rs72552713) is the most frequent with a MAF of 0.002 in the Asian population [108]. This mutation leads to an early stop codon; thus, no protein is produced. Several other, mainly nonsense mutations, e.g., R113X, R236X, R246X, G262X, E334X, and Q531X, result in a protein with severe structural and folding problems, and consequently in rapid protein degradation. Some other types of mutations, such as the deletion S340del, the frameshift L264Hfs, as well as the missense R147W, F208S, and R383C mutations [104,107], also belong to this group. A link between the lack of ABCG2 expression and the rare blood group Jr(a–) was established in 2012 [109,110]. Interestingly, individuals carrying Class 1 mutations on both alleles have no ABCG2 present in their RBC membranes, but exhibit no apparent signs of diseases. However, they can have transfusion reactions, and can be sensitive to drugs that are ABCG2 substrates, the concentration of which is normally controlled by the transporter.

### 4.4. Class 2 Mutations

Further embracing the CFTR classification, the next group of mutations (Class 2) consists of genetic alterations, usually mild mutations or polymorphisms, which result in a protein with impaired trafficking to the cell surface, while possessing more or less preserved transport function (recently reviewed in [137]). A typical of Class 2 variants is Q141K (rs2231142), which is the most frequent genetic variant of ABCG2 other than wild type. Its allele frequency is the highest in Southwest Asia (MAF values are between 0.222 and 0.319), whereas it is less common in the Caucasian populations (MAF = 0.107–0.119) and is rather rare in Africa (MAF = 0.009) [115,116]. Because of its frequency and its association with gout (see Section 6.1), ABCG2-Q141K is the most broadly studied variant of ABCG2. It has major folding and trafficking problems, undergoes partial degradation, thus exhibiting reduced overall and cell surface expression [32,33,95,100,115,117,118].

Using a flow cytometry-based screening method for reduced ABCG2 expression in RBCs, we have recently identified a polymorphic variant (M71V) with a character similar to that of Q141K [112,120]. The rare F373C missense mutation also belongs to this group, as the substrate stimulated ATPase activity and the specific transport activity of ABCG2-F373 are preserved, but its cell surface expression is diminished [104].

It should be noted that in most cases, not only the trafficking but also the transport function is affected by these mutations to certain extent. Nevertheless, they are classified into this group, because the defect caused by these mutations can—at least partially—be corrected by pharmacological chaperones (or correctors), whereas application of correctors for other types of mutations is usually unreasonable. The different approaches to rescue the various phenotypes will be discussed later.

### 4.5. Class 3 Mutations

The following group of ABCG2 mutations (Class 3) contains those that do not affect protein expression considerably, but impair transport function. At this point, the classification proposed here diverges from the categorization of the CFTR mutations, in which genetic alterations causing impaired gating are classified into Class 3, whereas those leading to diminished channel conductance are categorized into Class 4. This classification cannot be applied to ABCG2 for several reasons. Firstly, the transport mechanism of ABCG2 substantially differs from that of CFTR. Secondly, immense information has been accumulated on the mechanism of action of loss-of-function mutations in CFTR, whereas only limited data on ABCG2 mutations are available in this respect.

The genetic alterations in ABCG2 causing impaired transport activity are exemplified by S248P, P269S, F489L, and A528T [95,114,121,122,123]. The rare S476P also belongs to this group, although the transport function of ABCG2 bearing this mutation is only slightly reduced [104]. A recent addition to this class is I242T, which has normal protein expression and processing, while its transport activity (urate transport) is completely abolished [125]. A peculiar member of this group is K86M, an artificially generated mutation in the Walker A motif, which leads to a catalytic inactive transporter with close to normal cellular localization [9,126]. By means of these beneficial features, K86M is commonly employed in functional assays as a negative control.

It is worth noting that trafficking (Class 2) mutations cause reduced surface expression, and consequently lead to diminished drug efflux and increased drug sensitivity, to an apparent loss of function. Although generally speaking, the functionality of the transporter is affected in these cases, but the underling mechanism is rather different for Class 2 and Class 3 mutations. When a particular ABCG2 variant is evaluated solely on the basis of functionality in cellular systems, e.g., cell-based cytotoxicity or drug efflux assays, this distinction cannot be drawn. The various assessments to explore the particular impairments in the ABCG2 variants caused by mutations and polymorphisms will be discussed in Section 5. It is noteworthy that Class 3 mutations can potentially be corrected by so-called potentiators, small molecules that are intended to restore the impaired transport function (see Section 7).

### 4.6. Class 4 Mutations

As mentioned above, Class 4 of ABCG2 mutations differs from that of CFTR variants. According to our proposed classification, this group collects the mutations resulting in a transporter with altered substrate specificity. These mutations are represented by the rare F431L (rs750568956, MAF < 0.0001). ABCG2-F431L is normally expressed in cells, localized to the plasma membrane, mediates porphyrin transports, and confers resistance to mitoxantrone [95,113], but these cells exhibit decreased resistance to methotrexate and various tyrosine kinase inhibitors [95,138]. Conflicting results on altered resistance to camptothecin analog SN-38 (the active metabolite of irinotecan) have been published in connection with this mutant [95,128].

Other members of this class are the R482G and R482T missense mutations, which have been identified from drug-selected cell lines, when ABCG2 was originally cloned [4,5]. In the beginning, the actual sequence of the wild type ABCG2 was ambiguous; causing some controversies in the initial characterization of the transporter, but later it was made clear that there is an arginine at position 482 in the wt ABCG2. Amino acid alterations at R482 strongly influence the substrate profile of the transporter, which is exemplified by methotrexate or uric acid being a substrate of the wt ABCG2 but not of the R482G variant; and inversely, doxorubicin or daunorubicin is exported by ABCG2-R482G, but not by the wild type [3,86,129,130,131,132].

### 4.7. Class 5 Mutations

The next group of mutations consists of those that cause diminished protein expression (Class 5). Class 1 and Class 5 mutations differ from one another in respect of that the former results in no protein expression, whereas the latter only cause reduced protein levels, therefore, different strategies are required for rescuing the phenotype. A representative of this group is T153M (rs199753603), which leads to diminished protein expression, but the smaller amount of protein expressed normally traffics to the cell surface and functions regularly [104,107,128,133].

Although we focus here mostly on the mutations in the coding region, it is worth mentioning that SNPs in the promoter region or in introns (potentially causing alternative splicing) can influence RNA stability and consequently lead to diminished protein expression. These are exemplified by −30477C>G (rs2127861), −15622C>T (rs7699188), and 1143G>A (rs2622604) [134]. Since the transcriptional regulation and RNA splicing is tissue-specific, the manifestation of these SNPs can vary from tissue to tissue. It is noteworthy that correction of aberrant splicing by RNA-based antisense oligonucleotide strategy has been proposed to restore CFTR level caused by splice-altering mutations [139].

### 4.8. Class 6, Class 7, and Ambigous Mutations

Class 6 and Class 2 mutations are related, as both groups contain genetic alterations leading to trafficking defects, but while Class 2 mutations cause diminished plasma membrane delivery, Class 6 SNPs result in less stable protein on the cell surface. Reduced steady-state plasma membrane level of a membrane protein can be due to either enhanced internalization or diminished recycling back to the cell surface, which are often concomitant with augmented protein degradation. Unfortunately, only limited information is available on the half-life of various ABCG2 mutants on the cell surface; therefore, categorization to this class is rather vague at the moment. The rare missense mutation V441N (rs758900849, MAF < 0.0001) has been reported to cause reduced protein stability [95,100,122,127], thus likely be identified as a Class 6 mutation.

Recently, a new category (Class 7) has been proposed for CFTR mutations, which class collects the so-called unrescuable genetic alterations [97]. This phenotype can be due to a large deletion or other severe mutations generating unstable mRNA. The sparse information available at the moment prevents to classify ABCG2 mutations into this group.

Classification of certain mutations is ambiguous because of the conflicting results published on the particular defect caused by the SNPs. For instance, ABCG2 carrying the relatively frequent D620N (rs34783571 MAF = 0.003) or the rare N590Y (rs34264773, MAF = 0.0004) missense mutation exhibit normal or in some cases even elevated cell surface expression; however, whether these mutations affect the transport function remains to be clarified [107,113,114,135,136,140]. An interesting addition to list of ABCG2 SNPs is the rare I206L gain-of-function mutation (MAF = 0.0003) [135].

## 5. Assessment of Mutation-Related Defects in ABCG2

Numerous in vitro assay systems and cellular models have been established to evaluate the expression and function of various ABC transporters, including ABCG2. Most of these test systems are intended to identify substrate or inhibitor molecules of a given transporter. These drug-screening methods have been carefully overviewed in comprehensive review papers [141,142]. Here, rather, the approaches capable of identifying the impairments caused by various types of mutations/polymorphisms will be discussed.

As mentioned earlier, several ABCG2 SNPs are frequently miscategorized, as the mutated protein is characterized by one or just few features. For instance, Class 2 (trafficking) variants are often called loss-of-function mutants, although their transport function is in fact preserved. To obtain a comprehensive view on the character of an ABCG2 variant, numerous assessments have to be performed. These include the examination of gene transcription, mRNA level and stability, overall protein expression level, protein stability and degradation, cellular localization and trafficking, half-life in the plasma membrane, ATPase activities, transport function (the specific activity!), and substrate profile. How these various cellular parameters are affected by mutations in each class is summarized in Table 2.

The transcriptional activity of ABCG2 variants bearing mutations in the promoter region can be assessed by nuclear run-on or GRO-Seq (Global Run-On sequencing) assay; the combined outcome of transcription and RNA stability can be detected by RNA-Seq (RNA sequencing) method, whereas splice variants can be determined using RNase protection assay. A detailed analysis demonstrated differential regulation and cell-type specific appearance of four 5′ untranslated exon variants of ABCG2 [143]. Recently, a straightforward test system based on a genome-edited reporter cell was developed to assess transcriptional regulation of ABCG2 [144]. In these cells, a coding sequence for eGFP was targeted to the translational start site of ABCG2. This reporter cell can be adjusted to examine the transcription of ABCG2 variants.

Standard molecular biological and biochemical methods, such as quantitative PCR and Western blotting are used to determine the mRNA and protein levels, respectively. The subcellular localization of the ABCG2 variants is regularly analyzed by immunostaining followed by microscopy, or specifically the plasma membrane expression of ABCG2 can be assessed by cell surface labeling followed by flow cytometry. The commonly used cell surface labeling, however, leaves the question open as to whether the abnormal plasma membrane expression is due to reduced overall expression or trafficking deficiencies. It is also worth noting that no specific markers for subcellular compartments are employed in the majority of studies examining the subcellular localization of an ABCG2 variant, leaving its exact location within the cell unambiguous. In addition, it is frequently disregarded that the trafficking machinery differs from cell type to cell type; thus, application of close-to-physiologic cellular models is highly encouraged for localization studies. The limitation of these widely used immunolabeling approaches is that they provide information only on the steady-state distribution of the protein, and its end-point accumulation in a particular cellular compartment may not reveal the real cause of mislocalization. Recently, we developed a dynamic, synchronization-based method for identifying the specific impairments of Class 2 (trafficking) ABCG2 mutants, Q141K and M71V [120]. Similarly, a dynamic approach is needed to evaluate the protein half-lives on the cell surface, but these kinds of studies on ABCG2 are rather rare. Binding of the ABGC2-specific antibody 5D3 has been demonstrated to induce internalization [145], as well as binding of ABGC2-specific inhibitors has been shown to promote lysosomal degradation of the transporter [146]. However, no data on the internalization rates or the plasma membrane half-lives of different ABCG2 variants have been published thus far.

A wide variety of assay methods has been developed to evaluate the function and substrate recognition of ABCG2 variants (see in [141,142]). Some of these measurements are performed using membrane preparations or membrane vesicles containing the ABCG2 variant to be tested. Substrate molecules typically stimulate the ATPase activity of ABCG2 (like in many other ABC transporters), whereas inhibitors diminish the basal or substrate-stimulated activity. Compounds that modulate the function of the transporter, such as cholesterol, can also be identified using this method [147]. Photoaffinity labeling assay is also a membrane-based method, and indicates the interaction between a test compound and the transporter, while direct transport measurements can be performed employing inside-out membrane vesicles. These assays have the advantage that the specific activities can be determined in this way without being burdened by other factors like transcriptional differences, trafficking defects, or altered membrane half-lives. Moreover, using these approaches, substrate profiling for the different ABCG2 variants can be implemented.

Another group of functional assessments is cell-based methods, which employ cells expressing the ABCG2 variant to be tested. Cytotoxicity assay is the most commonly employed cell-based approach to investigate the transport activity of multidrug ABC transporters. The cell-killing effect of toxic compounds can be examined directly, but substances with no apparent toxic effect can be assayed through their modulatory effect on the cytotoxicity of a toxic drug. Measuring the efflux of a detectable substrate molecule from cells, or inversely, assessing its cellular accumulation (efflux and uptake assays) are other possibilities to determine the functional consequence of a mutation/polymorphism in ABCG2. Here too, the modulatory effect of other compounds on the cellular efflux or uptake of a detected substrate molecule allows mapping of potentially interacting substances. The so-called side-population assay is a representative of these cellular uptake methods [42]. The fluorescent dye Hoechst 33342 is a well-transported substrate of ABCG2, and its fluorescence undergoes a spectral shift upon binding to DNA. These properties make it possible to perform a dye exclusion assay for detecting a subset of cells with functional expression of ABCG2, e.g., stem cells, in a heterogeneous cell population by measuring cellular blue and red fluorescence in parallel [42,46,148,149]. Although cell-based assay systems are straightforward, and also allow substrate profiling, their limitation comes from the fact that the expression, localization, and cell surface stability of the tested ABCG2 variant indirectly affect the net outcome of these examinations.

It is also worth noting that in many heterologous cellular models, the transporter is considerably overexpressed, thus increasing the risk of experimental artefacts. Cell-based systems with a single or controlled number of transgene copies, such as Flp-In-293 system or transposon-based system, are more appropriate for such investigations [14,18,113,117,118,150]. For instance, an initial study using cellular models overexpressing ABCG2 reported that mutations at the glycosylation site (N596) do not affect trafficking of ABCG2 [12], but a subsequent study employing a single-copy Flp-In-293 system demonstrated a harmful effect of N596 mutations and established a stabilizing role for N-glycosylation [14]. Cellular models stably expressing the ABCG2 variant are commonly preferred over the transient systems, even though stable expression of a transgene allows us to investigate solely the steady-state distribution and function; furthermore, it may be accompanied with compensatory mechanisms. Transient and inducible expression systems should also be acknowledged, especially when dynamic cellular events, such as intracellular routing, are to be studied. Moreover, synchronous release methods, such as the RUSH (retention using selective hooks) system, are even more adequate for exploring the trafficking properties of different ABCG2 variants [120].

In addition to these ‘wet lab’ approaches, the effect of mutations and polymorphisms can be investigated by various in silico methods either by modelling their impact on the 3D structure of the transporter or by predicting their functional consequences, e.g., implicating an effect on protein stability, substrate binding, or intramolecular communication, etc. Although these are intriguing issues, this topic is beyond the scope of this review; thus, it not discussed here. In any case, structural analyses and molecular dynamic simulations in connection with ABCG2 mutations have recently been the subject of a comprehensive overview [30].

## 6. Medical Conditions Associated with ABCG2 Mutations and Polymorphisms

### 6.1. The Role of ABCG2 Variants in Hyperuricemia and Gout

One of the important roles of ABCG2 is the clearance of uric acid, the end-product of purine metabolism. Interestingly, uric acid was not identified as a physiologically relevant endogenous substrate of ABCG2 until genome-wide association (GWA) studies revealed the association between the Q141K polymorphism in the *ABCG2* gene and gout [151], a disease characterized by inflammatory arthritis of the joints caused by urate deposition and crystal formation in the synovial fluid. This condition can develop as a result of either hepatic overproduction or diminished excretion of uric acid. The Q141K polymorphism has been established as one of the strongest genetic determinants for high serum urate levels (hyperuricemia) and gout development [37,56,103,105,106,119,152]. Surprisingly, Q141K polymorphism (and its murine ortholog Q140K) diversely affected ABCG2 function in intestine and kidney, i.e., while ABCG2-mediated intestinal urate clearance was substantially reduced (or abolished), the function of ABCG2 was preserved in the kidney, suggesting a tissue-specific regulation/trafficking and differential pathology of this ABCG2 variant [58]. Ironically, another GWA study demonstrated an association between the Q141K polymorphism and a worse response to allopurinol [153], which is the first-line treatment for chronic gout. Allopurinol and its active metabolite, oxypurinol, are intended to lower uric acid levels by inhibiting xanthine oxidase, the major enzyme for the production of uric acid. Allopurinol and oxypurinol have been reported to be transported substrates of ABCG2; thus, they are retained in cells expressing the Q141K ABCG2 as compared to wt-expressing cells. A subsequent study demonstrated that ABCG2 transports oxypurinol, but not allopurinol itself [154]. Further pharmacogenomic analyses established the link between presence of the Q141K allele and increased risk of poor response to allopurinol [155,156,157], but the underlying mechanism, how Q141K influences allopurinol disposition and the clinical response, is yet to be clarified.

Several other ABCG2 mutations/polymorphisms have been identified as genetically associated risk factors for gout incidence (see Table 1). These include Q126X, R147W, T153M, and D620N [56,103,105,106,107]. No association between V12M and increased risk of gout have been established [56,103]; indeed, one study reported a protective effect of this SNP on gout susceptibility [105]. Recently, using blood samples from patients with hyperuricemia or gout, in addition to Q141K, we found three other mutations (M71V, R236X, and R383C) associated with reduced protein levels in the RBC membrane [112]. It is worth mentioning that most of these ABCG2 SNPs connected with hyperuricemia or gout belong to Class 1 or Class 2.

### 6.2. Modulatory Effect of ABCG2 Variants on Drug Pharmacokinetics

As discussed earlier, ABCG2 residing in various physiological barriers plays a pivotal role in tissue and cellular protection by controlling the uptake, distribution, and excretion of potentially toxic endogenous and exogenous substances. Mutations/polymorphisms affecting the functionality of the transporter by any means may also influence the serum level and/or the pharmacokinetic parameters of drugs that are ABCG2 substrates [96,158]. There are two consequences of altered pharmacokinetics: (i) the clinical response to a given drug can be modulated, and (ii) toxic reactions can arise due to shifted drug concentrations or drug–drug interactions. In particular, highly toxic drugs, like chemotherapeutic agents, can evoke adverse drug reaction (ADR) even at the usual dosage, when a polymorphism in transporters alters drug distribution. ABCG2 mutations/polymorphisms in connection with anti-cancer drug-elicited ADRs will be discussed in the following subsection. When drug concentrations (or exposures) are elevated due to the presence of a polymorphic variant, drug dosage should be adjusted accordingly to attain adequate response to treatment but to minimize side effects. However, compiling of dependable personalized treatment protocols demands not only reliable genetic analyses but also well-established knowledge of genotype–phenotype–pharmacokinetics relationships.

The effect of the frequent Q141K ABCG2 variant on drug pharmacokinetics has been extensively investigated, whereas only limited information has been acquired on other ABCG2 SNPs. Interestingly, the alterations in the disposition of several drugs correlate with the allele frequencies of this SNP in various ethnic groups [159]. Based on previous pharmacokinetic and pharmacodynamics data, a recommendation has been made to incorporate examination of ABCG2 Q141K in recent and future drug development [65]. Q141K has been demonstrated to modulate the pharmacokinetics of several types of drugs, including chemotherapeutic agents (discussed later), statins, disease-modifying anti-rheumatic drugs (DMARD), anticoagulants, and anti-viral medications. The pharmacokinetic parameters [the area under the curve (AUC) and/or the maximum concentration (C_max_)] of rosuvastatin, simvastatin, atorvastatin, and fluvastatin was elevated in patients carrying the Q141K allele [160,161,162,163,164]. Therefore, this polymorphism has also been implicated in increased the risk of statin-induced myopathy [165]. Q141K was also associated with higher AUC or C_max_ of DMARDs, such as sulfasalazine and teriflunomide [166,167,168,169], as well as of anti-HIV drugs, such as dolutegravir and retegravir [170,171]. It is worth mentioning that in many cases, association was seen only with the homozygous genotype, or with the group involving both hetero- and homozygotes. With regard to the other relatively frequent polymorphisms, V12M has demonstrated not to influence the pharmacokinetics of fluvastatin [162], whereas Q126X was associated with altered disposition of sulfasalazine [169].

Interestingly, a few studies have reported that the polymorphism-related alterations in pharmacokinetic parameters were primarily due to increased intestinal absorption rather than decreased renal excretion [159,171], although reduced hepatic clearance may also contribute to lower C_max_ values [166]. These notions are in concert with the differential pathology of Q141K in the intestine and the kidney observed in gout patients [58]. As discussed earlier, Q141K increases the risk of poor response to allopurinol by means of a not fully understood mechanism. However, further studies focusing on the oxypurinol pharmacokinetics and involving a large cohort of allopurinol-treated gout patients are needed to elucidate causative relationships.

### 6.3. Significance of ABCG2 Mutations/Polymorphisms in Cancer Therapy

Numerous in vitro studies have demonstrated altered efflux of chemotherapeutic agents and reduced drug resistance in cancer cells expressing ABCG2 carrying various SNPs. While an unambiguous role for Q141K has been established, the results on the effect of the V12M variant were conflicting [100,115,133,172]. It is important to note that the specific activity of both V12M and Q141K, i.e., transport activity normalized by the expression level, remained unaltered [100], confirming that these SNPs do not affect the actual transport function. ABCG2-Q141K has been shown to diminish drug efflux and/or to increase sensitivity to various anti-cancer drugs, including methotrexate, numerous TKIs (gefitinib, erlotinib, lapatinib, imatinib, dasatinib, nilotinib), and the topoisomerase I inhibitor indolocarbazole [100,114,115,133,172,173]. In addition to genetic alterations, epigenetic modifications could also modulate drug responses. It has been demonstrated that promoter hypermethylation, which frequently occurs in tumor cells, leads to repressed transcription of the *ABCG2* gene, lowered protein level, and consequently increased drug sensitivity [174].

An increasing number of clinical studies reported that ABCG2 polymorphisms modulate the pharmakinetics of chemotherapeutic drugs and increase the risk of drug-related adverse reactions. Studies rarely report altered drug response or clinical outcomes, but toxic side effects due to elevated drug concentrations or longer exposures. Higher accumulation of the small molecule TKI gefitinib was observed in patients heterozygous for the Q141K variant [81]. In accordance with this finding, higher incidence of gefitinib-induced diarrhea was found in Q141K heterozygotes [175], although another study found no association between susceptibility to gefitinib-induced ADR and the Q141K (or Q126X) polymorphism [176]. No difference was found in the therapeutic outcomes (response and survival) in diffuse large B-cell lymphoma patients carrying Q141K or V12M polymorphism, when treated with rituximab plus cyclophosphamide/doxorubicin/vincristine/prednisone (R-CHOP) regimen, but chemotherapy-induced diarrhea was associated with the Q141K genotype [177]. Similarly, a higher incidence of sunitinib-induced severe thrombocytopenia was observed in renal cell carcinoma (RCC) patients carrying the Q141K allele [178]. A case report also documented that an RCC patient homologous for Q141K suffered from toxic side effects of sunitinib treatment, such as severe thrombocytopenia, transaminase elevation, severe hypoxia due to pleural effusion and pulmonary edema, in parallel with efficacious treatment of the tumor [179]. Although the expression of ABCG2-Q141K in cancer cells resulted in elevated sensitivity to imatinib [114,173], the in vivo effect of Q141K on imatinib accumulation/response is controversial [180,181,182,183,184,185].

Similarly, the results on the role of this SNP in the pharmacokinetics of camptothecin analogs, such as diflomotecan, 9-aminocamptothecin, topotecan, and irinotecan, are conflicting [186,187,188,189,190,191,192]. An interesting dual role for the Q141K polymorphism was found in prostate cancer patients. On one hand, this ABCG2 variant results in elevated cellular retention of folate, which is a rate limiting factor for prostate cancer cell proliferation, thus leading to higher risk for tumor reoccurrence after prostatectomy in patients, who received no drug treatment. On the other hand, docetaxel-treated prostate cancer patients carrying Q141K SNP have a longer survival time because of the reduced drug efflux [193].

Besides Q141K, some other frequent polymorphisms, such as V12M and Q126X, have been included in a limited number of clinical studies investigating the effect of ABCG2 SNPs on cancer chemotherapy and therapy-related toxicity. V12M did not influence clinical outcomes and has no association with ADR; on the contrary, Q126X promoted toxic side effects of gefitinib [176,177,192]. Only sparse information is available on ABCG2 minor variants in connection with tumor drug therapy. The SNPs in the promoter region (-15622C/T) and in intron 1 (1143C/T) conferring reduced ABCG2 levels (Class 5 mutations) elevated pharmacokinetic parameters (AUC and C_max_) of erlotinib [194], and increased the incidence of gefitinib-induced diarrhea in non-small-cell lung cancer patients without affecting the clinical outcomes [195]. An SNP in intron 11 (G>A, rs4148157, MAF = 0.096) also altered the pharmacokinetics (absorption rate constant and maximum concentration) of orally administered topotecan in infants and very young children with brain tumors [196].

As mentioned earlier, ABCG2-inhibiting drugs can also modulate the distribution and toxicity of medications, underscoring the importance of ABCG2 in drug–drug or drug–food interactions. On the other hand, mutations and polymorphisms may potentially alter the potency and/or efficacy of these inhibitors, further complicating the evaluation of drug interactions. Nevertheless, this aspect of ABCG2 polymorphic variants remains to be determined.

### 6.4. Other Disease Conditions in Connection with ABCG2 Variants

Abcg2-deficent mice showed no signs of any noticeable phenotype until they were exposed to light, which in turn induced severe phototoxic lesions on the skin of the animals [67]. This led to the discovery that pheophorbide A, the breakdown product of chlorophyll, is an ABCG2 substrate, and the transporter restricts the intestinal uptake of this potential toxic compound. The level of the heme precursor protoporphyrin IX was also massively increased in the RBCs of Abcg2 knockout mice, implying a role for ABCG2 in heme homeostasis. This was further supported by the observations that ABCG2 is upregulated during erythroid differentiation and lowers the PPIX levels in erythroid cells [50]. A comprehensive in vitro study demonstrated that numerous ABCG2 polymorphisms, such as Q126X, F208S, S248P, E334X, S441N, and F489L, cause abrogated porphyrin transport regarding specific (normalized) activities [113]. It should, however, be noted that among these mutants, the expression level of the Class 1 variants (Q126X, F208S, and E334X) was practically zero, and thus no transport activity is expected anyway. These data and the observations with the Abcg2 knockout mice implicate a role for ABCG2 variants in erythropoietic protoporphyria (EPP), a disease of the heme biosynthesis pathway. EEP is caused by either genetic determinant (mutations in the heme biosynthesis enzymes), or by exposures to toxins or drugs, such as rifampicin and isoniazid. Accumulation of PPIX in EPP patients causes hepatotoxicity and phototoxicity primarily in the skin. A recent study elegantly demonstrated that Abcg2-deficency prevents mice from EPP-associated phototoxicity and hepatotoxicity by altering the disposition of PPIX [68]. In this case, this phototoxin is mostly retained in the RBCs, thus causing a reduced plasma level and preventing PPIX accumulation in the skin and the liver/bile. This surprising result clearly elucidates the multifunctionality of ABCG2: on one hand, it restricts the intestinal absorption of the xenobiotic phototoxin pheophorbide A, preventing light-induced skin damage; on the other hand, it facilitates release of the endogenous phototoxin PPIX from RBCs, contributing to the toxic effects, when PPIX is in excess (e.g., in EPP patients). Seeing pharmacogenomic or human in vivo data in connection with ABCG2 genotypes and ECC would be rather intriguing.

Several ABC transporters, including ABCA1, MDR1/ABCB1, MRP1/ABCC1, ABCG2, and ABCG4, have been implicated in the pathogenesis of Alzheimer’s disease (AD) [197,198], a progressive neurodegenerative disorder characterized by the deposition of amyloid-β (Aβ) peptides in the brain. Both in vitro and in vivo data demonstrated an ABCG2-dependent efflux of Aβ1-40, suggesting that ABCG2 at the blood-brain barrier prevents Aβ peptide from entering the brain [198,199]. In addition, an upregulation of ABCG2 has been found in the brains of AD patients with cerebral amyloid angiopathy [199], which was also reflected by increased ABCG2 expression levels observed in the RBCs of late-onset AD patients [69]. The role of ABCG2 in AD can be modulated by the mutations/polymorphisms. A lower prevalence of the Q141K was seen in late-onset AD patients, and interestingly increased susceptibility to AD was associated with the Q141K allele containing genotypes.

Besides AD, another neurodegenerative disorder, Parkinson’s disease (PD), has also been linked to ABCG2. Previously, higher levels of the natural antioxidant urate in the serum or the cerebrospinal fluid were associated with the clinical decline of PD patients [200,201]. Accordingly, the Q141K polymorphism was correlated with later disease onset of PD [202]; however, a recent study did not find association of the Q141K allele (or the high plasma level of urate) with the disease [203].

Intrauterine growth restriction (IUGR) is one of the most common forms of pregnancy complications. The idiopathic form of IUGR has also been linked to ABCG2, which is abundantly expressed in the placenta. Markedly decreased ABCG2 expression levels were found in placentas from IUGR pregnancies [204]. Based on this observation and the fact that trophoblasts in IUGR are subjected to excessive oxidative stress and apoptotic signals, a role for ABCG2 has been proposed in the protection of trophoblasts against stress-induced apoptosis, although the actual underlying mechanism is yet to be elucidated.

Recently, it has been reported that the expression level of ABCG2 is reduced in the RBC membranes of patients with type 2 diabetes and carrying the Q141K polymorphism, whereas this difference was not observed in patients homozygous for the wt ABCG2 [205]. The mechanism here too is yet to be clarified, but this observation clearly indicates the differential regulation of the wt and the Q141K polymorphic ABCG2 variants.

The actual role of ABCG2 in various stem cell types is still elusive, but it has become commonly accepted that ABCG2 has a protective role in stem cells, as they are exceptionally sensitive to environmental stresses [42,47,148,206]. The high-level expression of ABCG2 in pluripotent stem cells rapidly declines during differentiation [43], but may also regain in differentiated progeny cells that typically express ABCG2, e.g., in hepatocytes [207,208]. In addition to cell differentiation, various environmental impacts can elicit a drop in ABCG2 expression [43,44]. Not only its overall expression, but also its localization can be altered in response to stresses, e.g., mild oxidative stress evokes a reversible internalization of the transporter in pluripotent stem cells [209]. Nevertheless, little is known about the impact of various mutations/polymorphism in ABCG2 on its role in stem cell defense mechanisms. Exceptional medical relevance has the ABCG2 expression in cancer stem cells or drug-tolerant persisters, which can rapidly adapt to chemotherapy and repopulate the tumor [45,46,47,48,210]. The presence of these cell subpopulations in the tumors is an increasingly acknowledged reason for chemotherapy failures.

## 7. Efforts to Improve Impaired Trafficking or Function of ABCG2 Variants

Advancements in rescuing mutation-derived phenotypes of ABC transporters are largely driven by the research on CFTR, as cystic fibrosis is a frequent hereditary disease with adverse outcomes. The most frequent mutation in CFTR is F508del, which accounts for roughly 90% of all CF cases (~40–45% of the patients are homozygous for this mutation). F508del is considered to be a Class 2 mutation, since the majority of the protein bearing this mutation is retained in the endoplasmic reticulum (ER) by the ER quality control mechanism; nevertheless, the channel opening rate (gating) and the channel selectivity (equivalent to substrate specificity) are also affected by this mutation. Excessive efforts have been made to correct the various defects in CFTR, especially F508del [97]. A wide variety of molecules is employed to rescue not only CFTR mutants but also other deficient ABC transporters (reviewed in [211]).

Numerous small molecules can modulate the trafficking of membrane proteins. The inhibitors of cellular trafficking include fungal antibiotics, such as Brefeldin A, a blocker of the ER to Golgi transfer, Concanamycin A and Destruxin B, inhibitors of V-ATPases, as well as the immunosuppressant mycophenolic acid, a blocker of de novo GMP synthesis. Pharmaceutical drugs that are intended to restore trafficking and cell surface expression of the impaired transporter are named correctors, but are also called chemical or pharmacological chaperones, since most of these variants endure protein folding problems. The application of the correctors is reasonable for the phenotype rescue of Class 2 or conceivably Class 5 mutations. Corrector compounds include non-specific chemicals, such as glycerol or DMSO, but also small molecules with more specific effect. Corr-4a, Lumacaftor (VX-809), VRT-325, and recently Elexacaftor (VX-445) have been specifically developed to promote cell surface delivery of trafficking mutants of CFTR [212,213,214,215]. Surprisingly, Lumacaftor was effective in ameliorating cell surface delivery of trafficking-impaired variants of other ABC transporters, such as ABCA4 [216,217]. Moreover, VRT-325 has been shown to rescue the trafficking deficiency of ABCG2-Q141K, and restore urate efflux from cell expressing this ABCG2 variant [33].

The phenotype rescuing capability of the histone deacetylase inhibitor 4-phenylbutirate (4-PBA) has also been demonstrated for numerous ABC transporters, including CFTR [218], MDR1/ABCB1 [219], MDR3/ABCB4 [219,220], BSEP/ABCB11 [221], MRP6/ABCC6 [222], and ABCG2 [32,33,112]. The rescue effect of 4-PBA was attributed to stimulation of the expression of heat shock protein 70, a chaperon protein, which facilities folding of partially folded protein, thus preventing elimination by the ER-associated degradation (ERAD) system. Besides 4-PBA, several other histone deacetylase inhibitors, such as romidepsin, vorinostat, panobinostat, and valproic acid, have been demonstrated to restore cell surface expression of the mislocalized ABCG2-Q141K variant [118]. The underlying mechanism proposed here does not involve facilitated protein processing, but these histone deacetylase inhibitors have been implicated to block the dynein/microtubule retrograde transport, thus preventing the mutated ABCG2 from trafficking from the cell surface to aggresomes, the perinuclear vimentin-coated inclusion bodies [118]. Aggresomes are close to the centrosome and facilitate protein degradation by the autophagy pathway [223], and similar to ABCG2-Q141K, CFTR-F508del also tends to accumulate in this compartment [224]. In concert with the mechanism delineated above, colchicine, a blocker of microtubule polymerization, has been shown to elevate plasma membrane expression of ABCG2-Q141K by inhibiting its traffic to aggresomes [118]. This observation also has a medical relevance, since colchicine is a drug commonly used in gout therapy. It is worth also noting that the anti-cancer agent mitoxantrone also promote cellular processing of various ABCG2 variants [118,225], exemplified by Q141K, the accumulation of which in aggresomes was prevented by mitoxantrone [118].

Several types of ABCG2 defects involve protein degradation. Class 1 mutations cause major protein folding and stability problems; thus, ABCG2 variants carrying this type of mutations (e.g., Q126X) undergo rapid elimination by the ERAD system. Similarly, a fraction of Class 2 variants is subjected to degradative mechanisms; therefore, the overall steady-state protein expression levels of Class 2 variants (e.g., Q141K) are usually lower than that of the wild type. However, pharmacological chaperones can facilitate proper folding and save these variants from protein degradation. If the defect of a Class 5 variant is not due to diminished transcription or mRNA stability, but to reduced protein stability (e.g., S441N), it also undergoes at least partial degradation. Finally, Class 6 variants possessing shorter plasma membrane half-lives are also subjected to protein degradation, although they are primarily eradicated by the lysosomal system, whereas members of the Classes 1, 2, and 5 are rather eliminated by the proteasomal degradation. Most likely autophagosomal degradative mechanisms are also involved in the removal of deficient ABCG2 variants; however, little is known about contribution of this pathway. Blocking the various protein degradation mechanisms is also an option to restore reduced protein levels for these types ABCG2 mutants. According to the considerations above, the expression levels of Class 2 and Class 5 variants are likely to be improved by proteasome inhibition, whereas Class 6 variants can supposedly be corrected by blocking lysosomal degradation. However, this theoretical segregation does not entirely stand for tangible ABCG2 variants. For instance, the expression level of the ABCG-Q141K is increased 2-fold either by the proteasome inhibitor MG132 or by the lysosomal degradation blocker bafilomycin [118], indicating the involvement of both degradative pathways. Moreover, 3-methyladenine, an autophagosome inhibitor also cause a 3-fold increase in the expression level of the Q141K variant [118], implying a marked role for autophagosomal degradation in ABCG2 proteostasis. These findings, along with the observation that colchicine increased cell surface expression level of ABCG-Q141K suggest that a considerable fraction of these variant passes the central quality control mechanisms and reach the plasma membrane. This notion is further supported by the observation that, despite being subjected to proteasomal and autophagosomal degradations, the majority of the Q141K-ABCG2 variant is fully glycosylated, once having traversed the Golgi apparatus. This is in sharp contrast to what is observed with CFTR F508del or MDR1/ABCB1 mutants, which fail to become glycosylated and are entirely retained in the ER or targeted to proteasomal degradation [226,227,228]. It is worth noting that a large fraction of wt CFTR gets misfolded and consequently degraded [229,230], whereas wt ABCG2 hardly undergoes any proteasomal [14,112,225,231] or autophagosomal [118] degradation. The medical use of these inhibitors of protein degradation is, however, somewhat limited, as a general blocking of these vital cellular processes can be detrimental, despite the fact that proteasome inhibitors, such as MG132 or bortezomibe (Velcade), are used in cancer treatment, mainly as a component of combination therapies.

To restore the impaired transport function caused by Class 3 (and Class 4) mutations, so-called potentiator compounds are used. One of these small molecules is Ivacaftor (VX-770), which effectively restores the chloride channel activity of by loss-of-function CFTR mutants [213,214,232]. Although this drug has also been specifically developed for CFTR and is presumed to be selective for CFTR, surprisingly, it also rescues the phenotype caused by loss-of-function mutations in MDR3/ABCB4 [233], ABCA4 [216,217], and ABCB11 [234]. To our knowledge, no results on the correction of ABCG2 loss-of-function mutations by Ivacaftor have been published thus fur. The underlying mechanism is not quite understood, but Ivacaftor is presumed to somehow stabilize the structure of the impaired transporter. This drug has also been demonstrated to stimulate the ATPase activity of MDR1/ABCB1 [235], but most likely, this is not related to its rescuing effect, but rather indicates that Ivacaftor is a transported substrate. This notion is supported by the further observation that Ivacaftor competitively inhibited the MDR1-mediated transport of the fluorescent dye Hoechst 33,342 [235]. Nevertheless, further studies are needed to understand the potentiator-transporter interaction better and to elucidate the mechanism how Ivacaftor potentiates various ABC proteins.

## 8. Conclusions

Various mutations and polymorphisms in the *ABCG2* gene diversely affect the structure, stability, trafficking, and function of this multifunctional transporter protein. Gaining insight into the cellular fate of different ABCG2 variants promotes establishing specific interventions to overcome the medical condition caused by either the wild type or the mutated forms of ABCG2. The classification of ABCG2 mutations proposed here is based on the cellular features of various defects; thus, the variants categorized into the same class require similar kinds of efforts to rescue the phenotype. A broad and well-established knowledge base is a prerequisite for prudent therapies; thus, our better understanding of the genetics, biochemistry, and cell biology of ABCG2 variants bearing various mutations could help to establish effective personalized treatments that consider the patients’ genetic background.

## Figures and Tables

**Figure 1 ijms-22-02786-f001:**
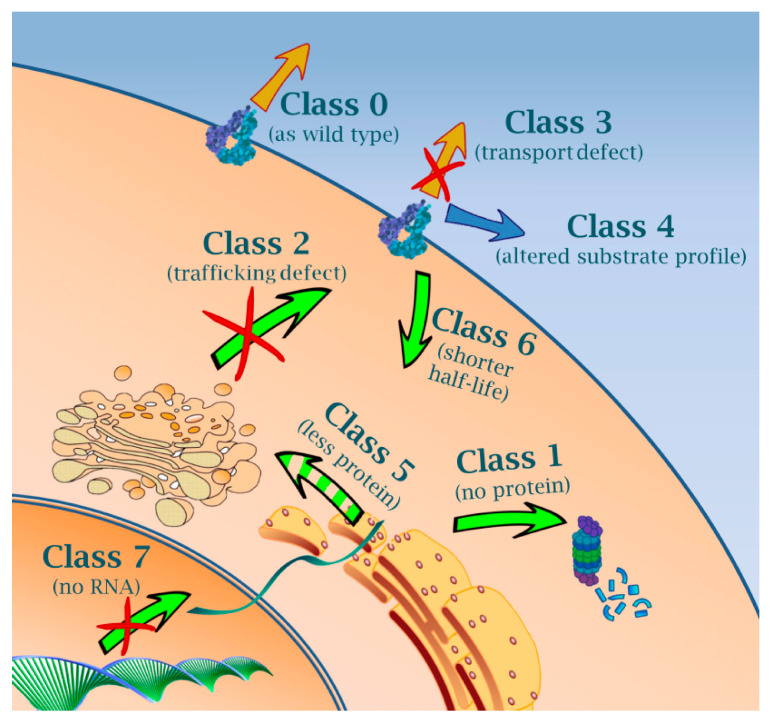
Classification of ABCG2 variants based on the cellular defects caused by the various mutations and polymorphisms. Green arrows—mRNA and protein trafficking, orange arrows—ABCG2-mediated transport, blue arrow—ABCG2-mediated transport with altered substrate preference.

**Table 1 ijms-22-02786-t001:** Proposed categorization of ABCG2 mutations and polymorphisms. Summary of the main features of the representative mutations of each group.

Class	Description	Variant	SNP Reference	Region	Global MAF	MAF in Asia	References	Assoc. with Gout	References for Gout Assoc.
Class 0	as wt	V12M	rs2231137	N terminal tail	0.158	0.19–0.33	[49,100,101,102]	dubious	no association in [56,103,105,106] but in [107]
		K360del	rs750972998	Linker	0.004	–	[103,104,107]	–	–
		T434M	rs769734146	TMH2	<10^−4^	–	[104,107]	–	–
Class 1	no protein	Q126X	rs72552713	NBD	0.0012	0.019	[108,109,110,111]	yes	[56,103,106]
		R236X	rs140207606	NBD	0.0002	0.0005	[49,104,109,110]	yes **	[112]
		R113X, Q244X, R246X, G262X, E334X, Q531X	miscellaneous	various locations	–	–	[109,110,113]	–	–
		S340del	rs755318857	Linker	<10^−4^	<10^−4^	[109]	–	–
		L264Hfs	rs780593948	NBD	<10^−4^	<10^−4^	[109]	–	–
		R147W	rs372192400	NBD	0.0001	–	[104,107]	yes	[107]
		F208S	rs1061018	NBD (Walker B)	<10^−4^	–	[95,111,114]	–	–
		R383C	–	Linker	–	–	[104]	yes **	[112]
Class 2	trafficking defect	Q141K	rs2231142	NBD	0.119	0.22–0.32	[32,33,95,100,115,116,117,118]	yes	[33,37,103,119]
		M71V	rs148475733	NBD	0.001	–	[104,112,120]	yes **	[112]
		F373C	rs752626614	Linker	<10^−4^	–	[104]		–
Class 3	reduced transport activity	S248P	rs3116448	Linker	<10^−4^	–	[95,121]	–	–
		S476P	not annotated	(CL1) TMH3	n.d.	–	[104,107]	–	–
		F489L	rs192169063	TMH3	0.001	0.005	[95,121,122,123]	–	–
		P269S	rs3116448	NBD:Linker	<10^−4^	–	[100,103,124]	no	[103]
		A528T	rs45605536	TMH4	0.02 *	–	[114]	–	–
		I242T	not annotated	NBD	–	–	[125]	–	–
		*K83M*	*–*	NBD, Walker A	–	–	[9,126]	*–*	*–*
Class 4	altered substrate recognition	F431L	rs750568956	TMH2	<10^−4^	–	[95,121,127,128]	–	–
		*R482G*	–	TMH3	–	–	[3,86,129,130,131,132]	–	–
		*R482T*	–	TMH3	–	–	[3,86,129,130,131,132]	–	–
Class 5	less protein	T153M	rs199753603	NBD:TM	0.0002	–	[104,107,128,133]	yes	[107]
		D296H	rs41282401	Linker	0.0002	0.02	[114]	–	–
		S441N	rs758900849	TMH2	<10^−4^	–	[95,100,122,127]	–	–
		L525R	rs750568956	TMH4	0.014	–	[114,122]	–	–
		−30477C>G	rs2127861	promoter	–	–	[134]	–	–
		−15622C>T	rs7699188	promoter	–	–	[134]	–	–
		1143G>A	rs2622604	intron 2	–	–	[134]	–	–
Class 6	shorter PM half-life	?	–	–	–	–	–	–	–
Class 7	no RNA	?	–	–	–	–	–	–	–
others	ambiguous	N590Y	rs34264773	EL3	0.0004	–	[101,128,135]	–	–
	ambiguous	D620N	rs34783571	EL3	0.003	–	[107,113,135,136]	yes	[107]
	gain-of- function	I206L	rs12721643	NBD (Walker B)	0.0003	–	[128,135]	–	–

PM, plasma membrane; NBD, nucleotide-binding domain; TMH, transmembrane helix; CL, cytoplasmic loop; EL, extracellular loop; n.d., no data; ?, unknown; * in Caucasians, ** observed in a small cohort of patients with hyperuricemia or gout.

**Table 2 ijms-22-02786-t002:** Different cellular parameters to be assessed, and their alterations caused by the mutations of various classes.

Cellular Parameter.	Class 0	Class 1	Class 2	Class 3	Class 4	Class 5	Class 6	Class 7
	as wt	No Protein	Trafficking Defect	Reduced Transport Activity	Altered Substrate Recognition	Less Protein	Shorter PM Half-Life	No RNA
Gene Transcription	+	+	+	+	+	+	+	+/−
mRNA Stability	+	+	+	+	+	+/−	+	+/−
mRNA Level	+	+	+	+	+	+/−	+	no
Protein Stability	+	reduced	+/−	+	+	reduced	+	N/A
Overall Protein Expression	+	no	+/−	+	+	reduced	+	N/A
Localization, Trafficking	normal	N/A	altered, impaired	normal	normal	normal	normal	N/A
Cell Surface Expression	normal	no	reduced	normal	normal	reduced	reduced	N/A
PM Half-Life	normal	N/A	N/A	normal	normal	normal	reduced	N/A
ATPase Activity	+	N/A	+	reduced	+	+	+	N/A
Transport (Specific Activity)	+	N/A	+	reduced	+	+	+	N/A
Substrate Profile	unchanged	N/A	+	+/−	altered	unchanged	unchanged	N/A

wt, wild type; PM, plasma membrane; +/−, normal or altered; N/A, not applicable. Color coding: green—not affected, normal, unchanged; red—altered, impaired; light red—can be altered; white—not applicable.

## Data Availability

Not applicable.

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
