# Peer review of "Medically Important Alterations in Transport Function and Trafficking of ABCG2"

_ijms, 2021, doi:10.3390/ijms22062786_

Round 1

Reviewer 1 Report

This is an interesting and informative review on the impact of genetic variants of the ABC transporter ABCG2 on the structure and function of this pump, as well as on the clinical repercussions of these alterations. In general, the paper is well written and easy to read.

Main points:

  1. Mentioning the crucial role of ABCG2 in bile acid transport across the placenta (PMID: 22096226) and mammary gland (PMID: 28785115), which may have clinical and pharmacological relevance (PMID: 24631341) in circumstances of maternal cholestasis, has been omitted. Accordingly, lines 220-222 must be corrected.
  2. An essential contribution of this review is the proposed classification of ABCG2 variants. I like it, but the order given to classify the variants seems capricious. I suggest re-ordering the seven (+0) Classes in a sequential manner based on the fate of the protein from mRNA to degradation.
  3. Figure 1. This scheme is useful to understand the classification of the ABCG2 variants, but this figure requires higher resolution.
  4. From line 317 to line 457, the description of different Classes should be divided into subsections (one Class one subsection) to make it easier to read.
  5. Line 905. National (?). The Country to whom the agency belongs must be specified here.

Author Response

1. Mentioning the crucial role of ABCG2 in bile acid transport across the placenta (PMID: 22096226) and mammary gland (PMID: 28785115), which may have clinical and pharmacological relevance (PMID: 24631341) in circumstances of maternal cholestasis, has been omitted. Accordingly, lines 220-222 must be corrected.

I greatly appreciate the reviewer’s thorough work and the positive evaluation of the manuscript. Indeed, the mentioned functions of ABCG2 and the associated pathology are relevant. This topic and the listed references have been included in the revised manuscript.

2. An essential contribution of this review is the proposed classification of ABCG2 variants. I like it, but the order given to classify the variants seems capricious. I suggest re-ordering the seven (+0) Classes in a sequential manner based on the fate of the protein from mRNA to degradation.

I absolutely agree that the order of mutation classes ranging from DNA/RNA through protein to degradation suggested by the reviewer would be more logical. However, the order I proposed in my manuscript follows the classification of CFTR mutations based on „theratypes”, which categorization has been used for many years. My opinion is that if we introduced a classification for ABCG2 mutations, which is more logical but differs from that for the CFTR, could cause confusion. Lots of efforts have been made by the ABC protein scientific community to introduce more or less uniform and consistent nomenclature, replacing the sometimes confusing traditional names (e.g., BCRP, MXR vs ABCG2). I believe that a consonant mutation classification between the various ABC protein subfamilies would be beneficial for the clarity and could help our better understanding. In the revised manuscript, I made efforts to make this point clearer.

3. Figure 1. This scheme is useful to understand the classification of the ABCG2 variants, but this figure requires higher resolution.

Thanks for the note. For the final version, a high quality image will be uploaded.

4. From line 317 to line 457, the description of different Classes should be divided into subsections (one Class one subsection) to make it easier to read.

Section 4 has has been divided into subsections in the revised manuscript.

5. Line 905. National (?). The Country to whom the agency belongs must be specified here.

The country has been specified.

  1.  

Reviewer 2 Report

Medically important alterations in transport function and trafficking of ABCG2

 László Homolya

The manuscript offers a comprehensive and up-to-date review of the ABCG2 transporter. With a focus primarily on transporter malfunction related to hyperuricemia and gout.

The author comes from a group with extensive experience in the subject matter, (Cells. 2019 Oct 8;8(10):1215; Cell Mol Life Sci. 2020 Jan;77(2):365-378) and the work is closely related to his recent publication: The ABCG2/BCRP transporter and its variants - from structure to pathology in FEBS Lett . 2020 Dec; 594(23):4012-4034.

The classification proposed to explain the cellular defects associated with the different mutations and polymorphisms of the transporter is a novel contribution (Figure 1 and Table 1) including some specific clarification in Table 2.

The last section exposes strategies to restore the trafficking of the variants describes investigations on CFTR and could be summarized.

On the other hand, the review mainly focused on gout could add relevant information   on polymorphism and transport to breast milk in breastfeeding women and the transport in term human placentas.

The physiological functions of should include that Riboflavin vitamin is actively transported into milk (van Herwaardenet al., 2007), and that it was specifically shown that mothers harboring the heterozygous c.421C > A polymorphism have a significantly higher milk to plasma nifedipine ratio than mothers harboring the c.421CC genotype.

line 9: hyperuricemia instead hyperureciema

Author Response

I am grateful for the reviewer for his/her thorough work and the positive opinion on the manuscript. In the revised manuscript a more detailed description on CFTR theratypes is given (see answer for Q2 of Reviewer 1).

On the other hand, the review mainly focused on gout could add relevant information   on polymorphism and transport to breast milk in breastfeeding women and the transport in term human placentas.

The physiological functions of should include that Riboflavin vitamin is actively transported into milk (van Herwaardenet al., 2007), and that it was specifically shown that mothers harboring the heterozygous c.421C > A polymorphism have a significantly higher milk to plasma nifedipine ratio than mothers harboring the c.421CC genotype.

Thanks for this note. In the revised manuscript, the physiological function of ABCG2 in the mammary gland and the placenta is detailed more. Also the connection with disease conditions are included (see answer for Q1 of Reviewer 1).

line 9: hyperuricemia instead hyperureciema

This typo has been corrected.